# Selective branch formation in ethylene polymerization to access precise ethylene-propylene copolymers

Yuxing Zhang[1,2], Xiaohui Kang [3✉] & Zhongbao Jian [1,2✉]

Polyolefins with branches produced by ethylene alone via chain walking are highly desired in industry. Selective branch formation from uncontrolled chain walking is a long-standing challenge to generate exclusively branched polyolefins, however. Here we report such desirable microstructures in ethylene polymerization by using sterically constrained α-diimine nickel(II)/palladium(II) catalysts at 30 °C–90 °C that fall into industrial conditions. Branched polyethylenes with exclusive branch pattern of methyl branches (99%) and notably selective branch distribution of 1,4-Me$_2$ unit (86%) can be generated. The ultrahigh degree of branching (>200 Me/1000 C) enables the well-defined product to mimic ethylene-propylene copolymers. More interestingly, branch distribution is predictable and computable by using a simple statistical model of $p(1-p)^n$ (p: the probability of branch formation). Mechanistic insights into the branch formation including branch pattern and branch distribution by an in-depth density functional theory (DFT) calculation are elucidated.

[1] State Key Laboratory of Polymer Physics and Chemistry, Changchun Institute of Applied Chemistry, Chinese Academy of Sciences, Changchun 130022, China. [2] University of Science and Technology of China, Hefei 230026, China. [3] College of Pharmacy, Dalian Medical University, Dalian 116044, China. ✉email: kangxh@dmu.edu.cn; zbjian@ciac.ac.cn

As the most important polymer by scale, polyolefins like polyethylene (PE) possess various architectures to form versatile products[1–3]. Branching is of great importance in these PEs to tune microstructures and thus determine material properties[4–9]. Via a traditional pathway, an exclusive branch in the PEs generally comes from a specific α-olefin, which is utilized as the comonomer in ethylene copolymerization. For instance, in the ethylene/propylene (EP) copolymer, linear low-density polyethylene (LLDPE), and polyolefin elastomer (POE) products, branch on the backbone of PEs is originated from $C_3$, $C_4$, $C_6$, or $C_8$ α-olefins. By comparison, the generation of branch from ethylene as the sole feedstock in the polymerization catalysis is highly attractive with regard to a simplified industrial process and the avoidable use of high-cost α-olefins[10,11].

Since Brookhart's seminal works in 1995[12–14] on α-diimine late transition metal such as nickel(II) and palladium(II) promoted olefin polymerization[15–21], a unique chain walking mechanism that involves a successive β-H elimination followed by a re-insertion event with an opposite regiochemistry has emerged as a powerful tool to generate branches from ethylene alone, allowing for the production of a variety of polyethylene topologies[4,22,23]. For instance, the architecture of polyethylene could readily be adjusted by a simple variation of ethylene pressure[4,24–26]. Over the past more than two decades, however, precise control on branches to produce well-defined polymer microstructures is a long-standing challenge, because of the uncontrolled chain walking process. A mixture of branch patterns including methyl branch and higher branches ($C_2$, $C_3$, $C_4$, and $C_{4+}$) predominantly occur (Fig. 1). Therefore, selective branch formation is difficult in ethylene polymerization via a chain walking manner. This is particularly challenging when the reaction temperature is high or the degree of branching is very high, which easily induces the formation of a mixture of branch patterns. In view of the huge difficulty on controlling branch pattern, branch distribution (namely interval between two adjacent branches) on the backbone of PE is more elusive thus far. These events evidently inhibit the synthesis of ethylene-propylene copolymers from ethylene at industrial temperatures of 40–70 °C[27].

We now report that the low-cost nickel(II) catalyst shows a superior control on selective branch formation in ethylene polymerization, enabling the production of an exclusive methyl branch pattern with an ultrahigh number in a particularly broad temperature range of 30–90 °C that meets the industrial process. Distribution of the formed branches in the obtained ethylene-propylene copolymers is highly selective based on a NMR analysis and is particularly predictable and computable based on a statistical model of probability. The underlying mechanism on the control of branch pattern and distribution is proposed and rationalized by an in-depth DFT calculation.

## Results

**Catalyst design and ethylene polymerization.** As a rationale of catalyst design, the desired nickel precursor ipty-Ni was readily synthesized from the reaction of $NiBr_2(DME)$ with the rigid and sterically constrained α-diimine ligand[28,29]. The structure and purity of ipty-Ni was identified by multiple techniques including $^1H$ NMR spectrum, elemental analysis, and X-ray diffraction analysis (Fig. 2; for detailed structural data, see Supplementary CIF file). As a comparison, ipty-Pd reported by us was also prepared (Fig. 2)[28], which produced highly branched polyethylenes with unsolved microstructures at low temperatures of <30 °C as well. With the activator of modified methylaluminoxane (MMAO), ipty-Ni exhibited high activities for ethylene polymerization to produce high to ultrahigh molecular weight polyethylenes (Table 1). Notably, the degree of branching was very high, which rose with elevating temperature from 30 to 90 °C and further increased to reach an ultrahigh level of 200 brs/1000 C (calculated from $^1H$ NMR data) with decreasing ethylene pressure from 8 bar to 2 bar[30,31]. Dependence of degree of branching on pressure and temperature is in agreement with previous results from a superficial analysis of $^1H$ NMR spectroscopy[32–35].

**Exclusive branch pattern and selective branch distribution.** To distinguish the branch pattern in these highly branched polyethylenes, instructive $^{13}C$ NMR spectra were in depth analyzed to provide insightful information (Table 2). Generally, high reaction temperature leads to the generation of long-chain branches, making selective branch more difficult. In contrast, whatever pressure (2 bar–8 bar) or temperature (30–90 °C) varies in our nickel system, the smallest methyl branches are exclusively (99%) detected in all polyethylene samples of Table 1 (for all $^{13}C$ NMR spectra, see Fig. 3). This is unprecedented that the branch pattern is unaltered and exclusive at such a high level of >200 brs/1000 C and at such high temperature of 90 °C. Importantly, the synthesis

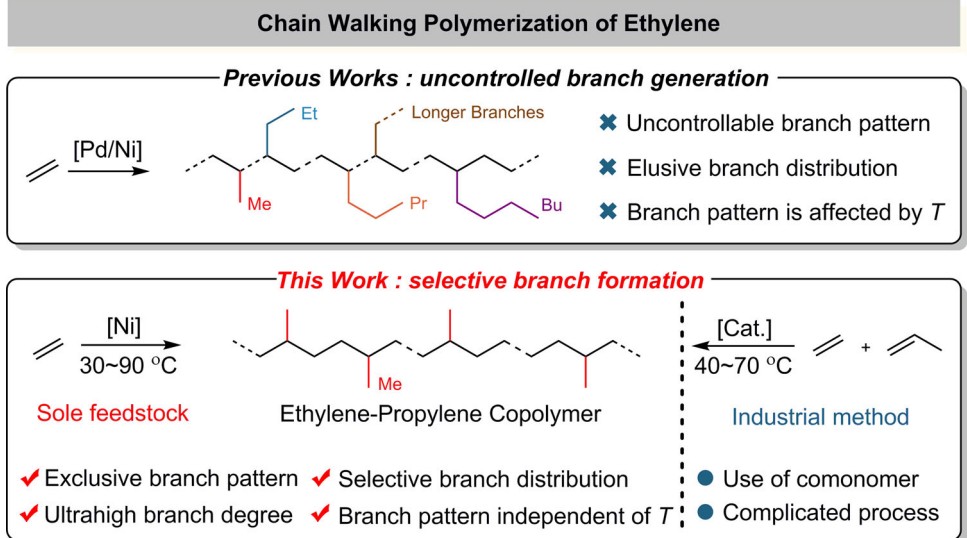

**Fig. 1 Feature of chain walking polymerization of ethylene.** A comparison on previous works and this work. Me: methyl; Et: ethyl; Pr: propyl; Bu: butyl; T: temperature.

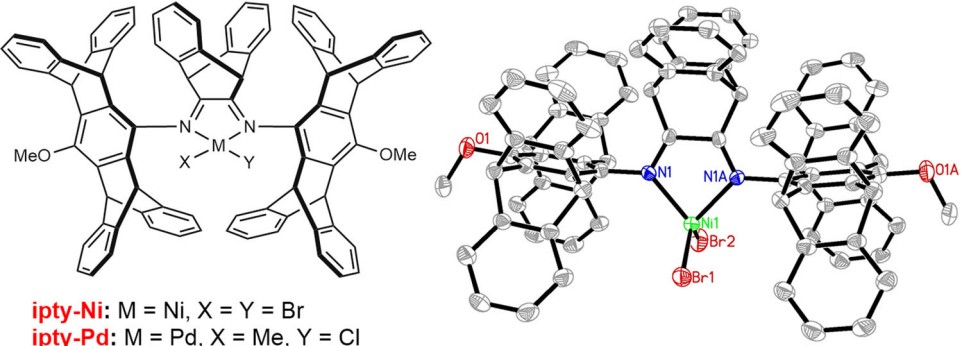

**ipty-Ni**: M = Ni, X = Y = Br
**ipty-Pd**: M = Pd, X = Me, Y = Cl

**Fig. 2 Sterically constrained catalysts of ipty-Ni and ipty-Pd and the molecular structure of ipty-Ni.** ipty-Ni and ipty-Pd are prepared by the reaction of α-diimine ligand with $NiBr_2$(DME) and PdMeCl(COD), respectively.

**Table 1 Effect of temperature and pressure on branching in ethylene polymerization with ipty-Ni[a].**

| Entry | $T$ (ºC) | $p$ (bar) | act. ($10^6$)[b] | $M_n$ ($10^4$)[c] | $M_w$ ($10^4$)[c] | $M_w/M_n$[c] | brs[d] |
|---|---|---|---|---|---|---|---|
| 1 | 30 | 8 | 2.70 | 129 | 166 | 1.29 | 101 |
| 2 | 50 | 8 | 2.58 | 86 | 128 | 1.49 | 139 |
| 3 | 70 | 8 | 2.55 | 65 | 103 | 1.60 | 168 |
| 4 | 90 | 8 | 1.53 | 47 | 72 | 1.52 | 179 |
| 5 | 90 | 6 | 1.02 | 36 | 59 | 1.65 | 188 |
| 6 | 90 | 4 | 0.54 | 35 | 53 | 1.50 | 197 |
| 7 | 90 | 2 | 0.30 | 18 | 33 | 1.84 | 200 |

[a]Reaction conditions: ipty-Ni (1 μmol), MMAO (500 eq.), toluene/$CH_2Cl_2$ (98 mL/2 mL), time (20 min), $T$ (temperature), $p$ (pressure). all entries are based on at least two runs, unless noted otherwise.
[b]Activity is in unit of $g\,mol^{-1}\,h^{-1}$.
[c]Determined by GPC in 1,2,4-trichlorobenzene at 150 °C using a light scattering detector.
[d]brs = Number of branches per 1000 C, as determined by $^1H$ NMR spectroscopy.

of ethylene-propylene copolymers from ethylene alone matches the industrial process of 40–70 °C[27], thanks to the ability of this nickel catalyst on controlling chain walking. Based on pioneering insights[4,12–14], high molecular weight of the obtained polyethylene implies the suppression of chain transfer that involves the replacement of the formed α-olefin by incoming ethylene. High degree of branching is attributed to high frequency of β-H elimination and subsequent re-insertion. Notably, the key exclusive formation of methyl branches means that 2,1-insertion of α-olefins predominates and no event occurs from 2,1-insertion of internal olefin that generates longer branches.

The exclusive methyl branches further allow for an in-depth analysis on branch distribution via $^{13}C$ NMR spectra[36–39]. As depicted in Fig. 3 (205 Me/1000 C), extremely low signal intensities in the range of 10.0–18.0 ppm corroborate trace amounts of long-chain branches and 1,2-$Me_2$ unit. By combining previously reported resonances with DEPT-135 and 2D NMR spectra, 1,4-$Me_2$ unit, 1,6-$Me_2$ unit, and 1,x-$Me_2$ units (x = 8, 10, 12…) are unambiguously identified. Most notably, the content of both 1,4-$Me_2$ unit and 1,6-$Me_2$ unit can be calculated as 70.5% and 21.0%, respectively, according to the quantitative $^{13}C$ NMR spectrum. However, beyond the resolution of NMR technique in the experimental scale, 1,x-$Me_2$ units (x = 8, 10, 12…) are hardly distinguished due to the unidentified long $(CH_2)_n$ units.

**Proposed mechanism and computable branch distribution**. To address this issue, we propose a mechanism for the formation of branch pattern and branch distribution (Fig. 4), which will be comprehensively discussed in the DFT calculation section (see below)[40]. Preliminarily, two distinct features should be noted:

distribution of two neighboring methyl branches separated by an odd number of methylene ($CH_2$) group is impossible, this means 1,3-$Me_2$ unit, 1,5-$Me_2$ unit, and 1,(x-1)-$Me_2$ units (x = 8, 10, 12…) cannot be generated via a chain walking manner in ethylene polymerization; on the other hand, the formation of 1,2-$Me_2$ unit and long-chain branches such as 1,(x-1)-Me/Et units (x = 4, 6, 8, 10, 12…) is proposed, despite trace amounts.

In terms of branch distribution, we envision a statistical model of probability. We set up a simple calculation (Table 3, and for details, see Supplementary Excel file for statistic model) that assumes on each step of the proposed scheme (Fig. 4) either formation of a branch occurs (with probability p) or chain growth occurs (with probability 1-p). That is the 1,4-$Me_2$ unit structure is formed by a branch formation (probability: p), the 1,6-$Me_2$ unit structure is formed by one growth step and a branch formation [probability: p(1-p)], and so on. Notably, this model that assumes the probability of branch formation vs. growth is independent of the distance from the previous branch. This calculated data based on the model agrees with experimental quantitative $^{13}C$ NMR data quite well (see branch distribution in Table 3). For example, in the Supplementary Excel file we set with p = 70.5% of 1,4-$Me_2$ unit (type into field B1 as 0.705), and then 20.8% of 1,6-$Me_2$ unit (21.0% from NMR data), 6.1% of 1,8-$Me_2$ unit, 1.8% of 1,10-$Me_2$ unit, 0.5% of 1,12-$Me_2$ unit and more units are automatically generated (also input the other p value in Supplementary Excel).

This advanced method solves the shortcoming of NMR technique and enables branch distribution predictable and computable, where the ratio of (1,4)-α, [(1,6)-α+(1,x)-α], (1,6)-β, (1,x)-β, (1,x)-γ, (1,x)-δ, CH, and Me signals in the statistical model set by us well fits in the calculated value from $^{13}C$ NMR data in Fig. 3. As an extension, branch number (brs) is also easily calculated and predicted by setting an equation of $I_{[Me\ branches]}/I_{[all\ signals]}$ (entry 7: 205/1000 C ($^{13}C$ NMR) vs 207/1000 C (Model)) (Supplementary Excel file).

These highly branched polyethylenes with exclusive methyl branches produced by the palladium precursor ipty-Pd were further analyzed with regard to branch distribution (Table 2, entries 8–10). At 0 °C, the selectivity of methyl branch pattern is 99%, and most notably the selectivity of 1,4-$Me_2$ branch distribution reaches the highest value of 86%. As anticipated, distribution of these branches is also computable. Note that long-chain branches generate at elevated temperatures of >30 °C in palladium species, indicating a reduced control on chain walking relative to nickel species. To the best of our knowledge, in chain walking polymerization selective branch pattern of methyl group generally occurs in the α-olefin reaction via a 1,2-insertion and subsequent ω, 2-enchainment[41–44]; however, highly selective 1,4-$Me_2$ branch distribution is thus far only generated in 1-butene or *trans*-2-butene reactions at low temperatures such as −40 °C[45–49].

**Table 2 Analysis of branch pattern and distribution.**

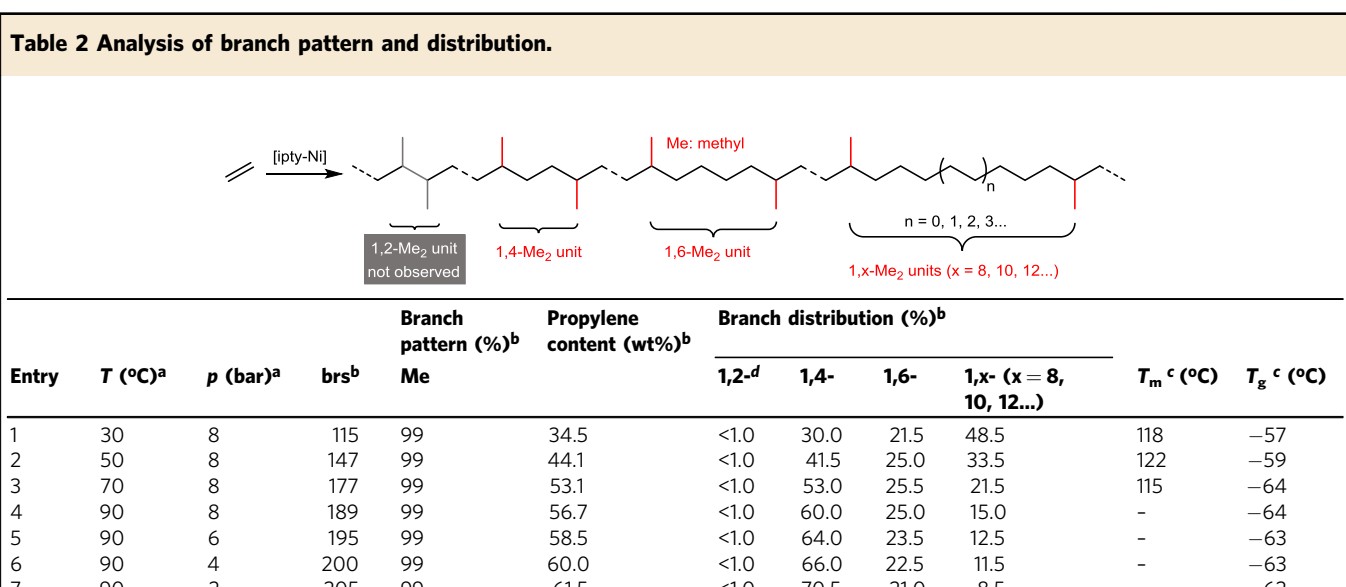

| Entry | T (°C)[a] | p (bar)[a] | brs[b] | Branch pattern (%)[b] Me | Propylene content (wt%)[b] | Branch distribution (%)[b] | | | | $T_m$[c] (°C) | $T_g$[c] (°C) |
|---|---|---|---|---|---|---|---|---|---|---|---|
| | | | | | | 1,2-[d] | 1,4- | 1,6- | 1,x- (x = 8, 10, 12...) | | |
| 1 | 30 | 8 | 115 | 99 | 34.5 | <1.0 | 30.0 | 21.5 | 48.5 | 118 | −57 |
| 2 | 50 | 8 | 147 | 99 | 44.1 | <1.0 | 41.5 | 25.0 | 33.5 | 122 | −59 |
| 3 | 70 | 8 | 177 | 99 | 53.1 | <1.0 | 53.0 | 25.5 | 21.5 | 115 | −64 |
| 4 | 90 | 8 | 189 | 99 | 56.7 | <1.0 | 60.0 | 25.0 | 15.0 | – | −64 |
| 5 | 90 | 6 | 195 | 99 | 58.5 | <1.0 | 64.0 | 23.5 | 12.5 | – | −63 |
| 6 | 90 | 4 | 200 | 99 | 60.0 | <1.0 | 66.0 | 22.5 | 11.5 | – | −63 |
| 7 | 90 | 2 | 205 | 99 | 61.5 | <1.0 | 70.5 | 21.0 | 8.5 | – | −62 |
| 8[e] | 0 | 8 | 230 | 99 | 69.0 | <1.0 | 86.0 | 12.5 | 1.5 | – | −61 |
| 9[e] | 15 | 8 | 224 | 99 | 67.2 | <1.0 | 80.0 | 16.5 | 3.5 | – | −60 |
| 10[e] | 30 | 8 | 213 | 98 | 63.9 | <1.0 | 73.0 | 20.5 | 6.5 | – | −62 |

[a]$T$ (temperature), $p$ (pressure).
[b]brs = Number of branches per 1000 C, as determined by $^{13}$C NMR spectroscopy.
[c]Determined by DSC (second heating).
[d]Actually, the total content of 1,2-Me$_2$ unit of Methyl branch and the other patterns of branches is less than 1.0%.
[e]Polymers produced by ipty-Pd in ref. [28] (Note: the previously reported branching degrees of 220, 218, 201/1000 C are calculated according to $^1$H NMR spectra. A small deviation on the degree of branching is normal between $^1$H NMR spectroscopy and $^{13}$C NMR spectroscopy).

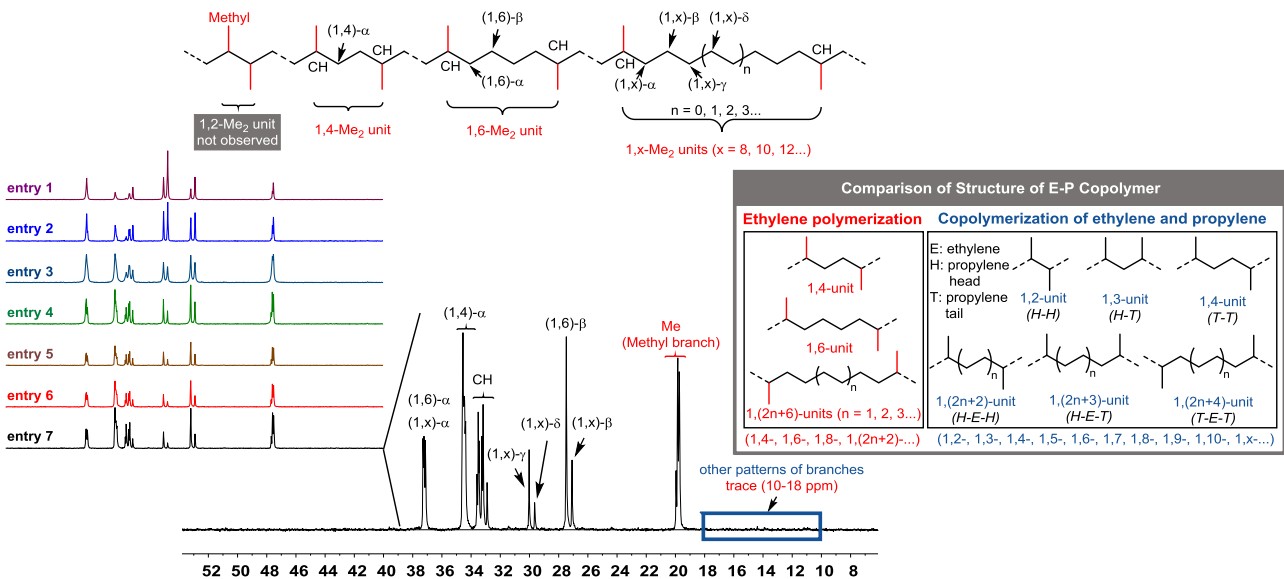

**Fig. 3 $^{13}$C NMR spectra and an in-depth analysis of branch pattern and distribution.** $^{13}$C NMR spectra (400 MHz, C$_2$D$_2$Cl$_4$, 110 °C) of highly branched polyethylenes come from Table 1, entire 1–7 and the in-depth analysis comes from Table 1, entry 7 (205 branches/1000 C). Comparison on the microstructure of ethylene-propylene copolymer produced either by ethylene polymerization or by copolymerization of ethylene (E) and propylene (P).

Focusing on the microstructure of these ethylene-propylene copolymers produced by ethylene polymerization, only 1,4-, 1,6-, 1,8-, and 1,(2n + 2)- (n = 1, 2, 3…) units with an even interval between two neighboring methyl branches are observed in chain walking polymerization with ethylene as the sole monomer. As a comparison, the microstructure of ethylene-propylene copolymer produced by copolymerization of ethylene and propylene is distinct (Fig. 3). 1,2-, 1,3-, 1,4-, 1,5-, 1,x- (x = 2, 3, 4…) units generated by *H-H*, *H-T*, *T-T* linkages and *H-E-H*, *H-E-T*, *T-E-T* linkages between ethylene and propylene [36].

**Mechanistic insight into branch formation (nickel system).** For more detailed discussion, Cf. Supplementary Notes, and Figs. 1–10. All structures of transition states and intermediates can be found in Supplementary coordinates.XYZ file. The intriguing branching character stimulated us to study the related mechanism of ethylene polymerization by nickel species ipty-Ni through using DFT calculations (Fig. 5). At first, the reaction starts from the β-H elimination (BHE) based on a β-agostic species $1_{β\text{-}T}$ to give a propylene-coordinated complex **1**. This process, which overcomes a free energy barrier (**TS$_{1BHE}$**) of 12.9 kcal mol$^{-1}$, is endergonic by

**Fig. 4 Proposed mechanism for the formation of branch.** Possible branch pattern and distribution formed by chain walking in ethylene polymerization.

**Table 3 Precise analysis of branch distributions by a statistical model of probability.**

| p = probability of formation of branch | | | | Branch distribution (%) of methyl units | | | | | |
|---|---|---|---|---|---|---|---|---|---|
| 1-p = probability of chain growth | | | | 1,4- | 1,6- | 1,8- | 1,10- | 1,12- | 1,(2n + 4)- [n ≥ 5] |
| Entry 1 | p = 30.0% | Experimental data | | 30.0 | 21.5 | Cannot distinguish by NMR technique | | | |
| | | Calculated data | | 30.0 | 21.0 | 14.7 | 10.3 | 7.2 | See Supplementary Excel |
| Entry 2 | p = 41.5% | Experimental data | | 41.5 | 25.0 | – | | | |
| | | Calculated data | | 41.5 | 24.3 | 14.2 | 8.3 | 4.9 | See Supplementary Excel |
| Entry 3 | p = 53.0% | Experimental data | | 53.0 | 25.5 | – | | | |
| | | Calculated data | | 53.0 | 24.9 | 11.7 | 5.5 | 2.6 | See Supplementary Excel |
| Entry 4 | p = 60.0% | Experimental data | | 60.0 | 25.0 | – | | | |
| | | Calculated data | | 60.0 | 24.0 | 9.6 | 3.8 | 1.5 | See Supplementary Excel |
| Entry 5 | p = 64.0% | Experimental data | | 64.0 | 23.5 | – | | | |
| | | Calculated data | | 64.0 | 23.0 | 8.3 | 3.0 | 1.1 | See Supplementary Excel |
| Entry 6 | p = 66.0% | Experimental data | | 66.0 | 22.5 | – | | | |
| | | Calculated data | | 66.0 | 22.4 | 7.6 | 2.6 | 0.9 | See Supplementary Excel |
| Entry 7 | p = 70.5% | Experimental data | | 70.5 | 21.0 | – | | | |
| | | Calculated data | | 70.5 | 20.8 | 6.1 | 1.8 | 0.5 | See Supplementary Excel |

Branch formation branching tree: Growth → p → 1,4; 1-p → p(1-p)$^1$ → 1,6; 1-p → p(1-p)$^2$ → 1,8; 1-p → p(1-p)$^3$ → 1,10; 1-p → p(1-p)$^4$ → 1,12; 1-p → p(1-p)$^n$ → 1,2n+4

9.5 kcal mol$^{-1}$. This unstable **1** goes through a slightly lower **TS$_{12reins}$** with an energy barrier of 12.0 kcal mol$^{-1}$ to complete propylene re-insertion with a 2,1-manner, generating a new β-agostic intermediate **2**. By comparison, **2** is the more both kinetically and thermodynamically favorable than **1$_{β-T}$**. Therefore, a new ethylene coordination separating β-agostic-H based on **2** is considered, and is found to be feasible via an energy barrier of 12.7 kcal mol$^{-1}$, leading to **3**. Finally, an ethylene insertion into the Ni−isopropyl bond of **3** occurs via a **TS$_{34ins}$** with an energy barrier of 10.0 kcal mol$^{-1}$ to generate a methyl-branching **4**, accompanied with an energy release of 15.1 kcal mol$^{-1}$.

Starting from **4**, two possible pathways, viz., the direct ethylene coordination−insertion (**4** → **TS$_{45coor}$** → **5** → **TS$_{56ins}$** → **6**) and the BHE−2,1-reinsertion (**4** → **TS$_{5BHE}$** → **5$_{BHE-C}$** → **TS$_{56reins}$** → **6$_{reins}$**),

were calculated. The former pathway has a total energy barrier of 14.2 (ΔG$_1$$^‡$, −3.4−(−17.6), **TS$_{45coor}$**) kcal mol$^{-1}$ and is exergonic by 11.4 (−29.0−(−17.6)) kcal mol$^{-1}$. By contrast, a higher energy barrier of 18.1 (ΔG$_2$$^‡$, 0.5−(−17.6), **TS$_{56reins}$**) kcal mol$^{-1}$ and an endergonic character (5.0 = −12.6−(−17.6), **6$_{reins}$**) in the latter case suggest that 1,2-Me$_2$ unit (**6$_{reins}$**) is inaccessible in both kinetics and thermodynamics, which is in line with the experimental observation. Subsequent ethylene insertion into **6$_{reins}$** (**6$_{reins}$** → **TS'$_{67coor}$** → **7'** → **TS'$_{78ins}$** → **8'**) has a higher energy barrier of 21.9 (ΔG$_3$$^‡$, 4.3−(−17.6), **TS'$_{67coor}$**) kcal mol$^{-1}$, further confirming that the reverse reaction of 1,2-Me$_2$ unit formation more easily occurs.

To further theoretically access the origin of selectivity on exclusive methyl branches, that is why 1,4-Me$_2$ unit can be produced but longer-chain branch units such as 1,3-Me/Et unit are

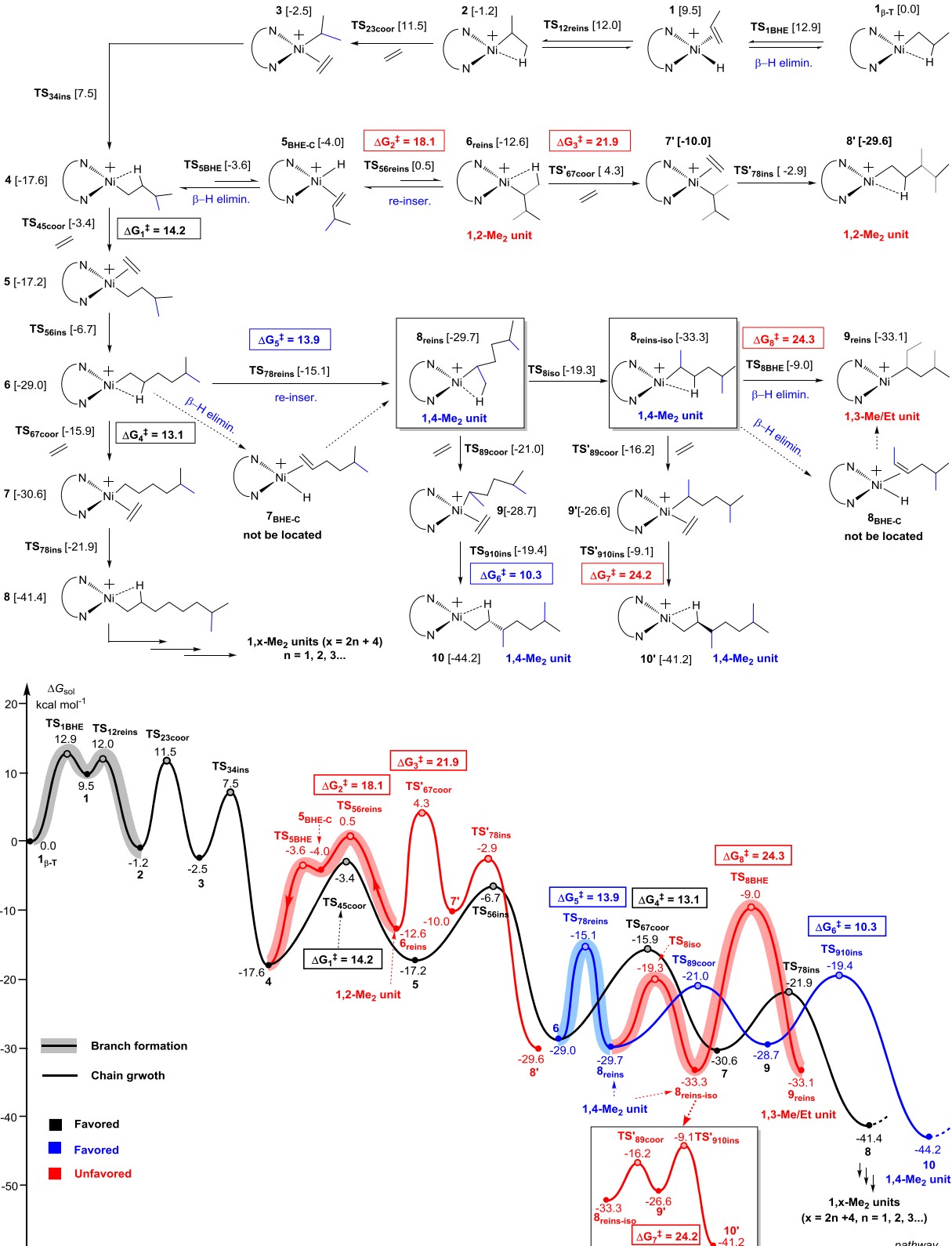

**Fig. 5 Mechanism for the formation of branch generated by ipty-Ni from the DFT simulation.** The mechanism involves the origin of selectivity, namely the inaccessible 1,2-Me$_2$ unit (distribution) and 1,3-Me/Et unit (longer branches) and the available 1,4-Me$_2$ unit. TS: transition state; BHE: β-H elimination; elimin.: elimination; reins: reinsertion; re-inser.: re-insertion; coor: coordination; ins: insertion; iso: isomerization; reins-iso: reinsertion-isomerization; BHE-C: β-H elimination-complex.

unavailable, we also investigated further ethylene insertion and BHE based on **6**. As indicated in Fig. 5, the direct ethylene coordination-insertion overcomes a total energy barrier of 13.1 ($\Delta G_4^{\ddagger}$, $-15.9-(-29.0)$, **TS$_{67coor}$**) kcal mol$^{-1}$ and releases an energy of 12.4 ($-41.4-(-29.0)$) kcal mol$^{-1}$, yielding a precursor (**8**) of 1,x-Me$_2$ units (x = 2n + 4, n = 1, 2, 3…). To obtain 1,4-Me$_2$ unit and 1,3-Me/Et unit, a BHE and 2,1-reinsertion from **6** was calculated. Notably, locating 5-methyl-1-hexene-coordinated intermediate **7$_{BHE-C}$** is fruitless, and just one transition state **TS$_{78reins}$** with an energy barrier of 13.9 ($\Delta G_5^{\ddagger}$, $-15.1-(-29.0)$) kcal mol$^{-1}$ is obtained between **6** and **8$_{reins}$** (1,4-Me$_2$ unit, observed in the experiment) (Detailed discussion for the not located **7$_{BHE-C}$** in Supplementary Figs. 5 and 6). Then, **8$_{reins}$** ($-29.7$ kcal mol$^{-1}$, the β-agostic H from CH$_3$ of methyl branches) isomerizes to a more stable **8$_{reins-iso}$** ($-33.3$ kcal mol$^{-1}$, the β-agostic H from CH$_2$ of backbone, also 1,4-Me$_2$ unit). It is noteworthy that the generations of 1,6-Me$_2$ unit precursor ($\Delta G_4^{\ddagger}$ = 13.1 kcal mol$^{-1}$) and 1,4-Me$_2$ unit ($\Delta G_5^{\ddagger}$ = 13.9 kcal mol$^{-1}$) indicate the similar energy barriers, suggesting that they both can be obtained. This shows a good agreement with experimental results.

Furthermore, in following BHE and reinsertion from **8$_{reins-iso}$** to **9$_{reins}$** (1,3-Me/Et unit), we cannot also locate 5-methyl-2-hexene (MH)-coordinated intermediate **8$_{BHE-C}$** and just obtain one transition state **TS$_{8BHE}$** (Supplementary Figs. 7 and 8). Unlike 1,4-Me$_2$ unit, the formation of 1,3-Me/Et unit needs going through a higher **TS$_{8BHE}$** with an energy barrier of 24.3 ($\Delta G_8^{\ddagger}$, $-9.0-(-33.3)$) kcal mol$^{-1}$ and is an apparently unfavorable process, which is consistent with experimental findings.

Beyond those, the possibilities of further polymerization based on **8$_{reins}$** and **8$_{reins-iso}$** were also considered. It was found that the ethylene coordination-insertion (**8$_{reins}$** → **TS$_{89coor}$** → **9** → **TS$_{910ins}$** → **10**) into **8$_{reins}$** owns an energy barrier of 10.3 ($\Delta G_6^{\ddagger}$, $-19.4-(-29.7)$) kcal mol$^{-1}$ and is exergonic by 14.5 ($(-44.2-(-29.7))$) kcal mol$^{-1}$, while the process (**8$_{reins-iso}$** → **TS'$_{89coor}$** → **9'** → **TS'$_{910ins}$** → **10'**) involving **8$_{reins-iso}$** overcomes a higher energy barrier of 24.2 ($\Delta G_7^{\ddagger}$, $-9.1-(-33.3)$) kcal mol$^{-1}$ and is a less exergonic process ($-7.9 = -41.2-(-33.3)$ kcal mol$^{-1}$) (Detailed discussion for **TS$_{910ins}$** and **TS'$_{910ins}$** in Supplementary Fig. 9). Therefore, the subsequent insertion on the basis of **8$_{reins}$** is smooth. Aforementioned findings allow us to conclude that the unique ancillary ligand forms a suitable channel for an exclusive formation of small methyl branches and a precise distribution of 1,2n + 4-Me$_2$ units (n = 0, 1, 2, 3…) in highly branched polyethylenes.

## Mechanistic insight into branch formation (difference of nickel and palladium system).

The entire discussion and the related Supplementary Notes and Fig. 3 and 10 can be found in Supplementary Information. Herein, we only focus on the difference of the origin of selectivity based on the same unique ligand. Inaccessible 1,2-Me$_2$ unit (distribution): (**Ni**) the unfavorable kinetics ($+3.9$ kcal mol$^{-1}$) and thermodynamics ($+5.0$ kcal mol$^{-1}$) and high barrier (21.9 kcal mol$^{-1}$) of subsequent ethylene insertion; (**Pd**) the favorable kinetics ($-5.8$ kcal mol$^{-1}$) but both unfavorable thermodynamics ($+2.3$ kcal mol$^{-1}$) and high barrier (17.8 kcal mol$^{-1}$) of subsequent ethylene insertion. Inaccessible 1,3-Me/Et unit (longer branches): (**Ni**) the extremely unfavorable kinetics ($+14.0$ kcal mol$^{-1}$); (**Pd**) the slightly unfavorable kinetics ($+2.2$ kcal mol$^{-1}$), indicating an impossibility at low temperature but a possibility at higher temperature. Available 1,4-Me$_2$ unit: (**Ni**) comparable barrier of isomerization (chain walking) with the precursor of 1,6-Me$_2$ unit (13.9 vs 13.1 kcal mol$^{-1}$) but lower barrier (10.3 kcal mol$^{-1}$) of subsequent ethylene insertion to form 1,4-Me$_2$ unit (**10**); (**Pd**) easy isomerization (chain walking) (7.3 kcal mol$^{-1}$) and comparable

barrier (15.0 vs 15.5 kcal mol$^{-1}$) of subsequent ethylene insertion to form 1,4-Me$_2$ unit (**10**).

## Discussion

In summary, we have demonstrated that challenging selective branch formation in ethylene polymerization via a chain walking manner is now achieved by a sterically constrained nickel catalyst at broad reaction temperatures (also by the corresponding palladium analog at low temperatures). This overcomes the general propensity of complex microstructures produced by uncontrollably successive β-H elimination and opposite re-insertion events. Thus, an exclusive pattern of ultrahigh methyl branches is accessible, along with a highly selective branch distribution of 1,4-Me$_2$ unit. Notably, branch distribution is unprecedentedly predictable and can be easily calculated by a simple statistical model of probability. The obtained polymers with methyl branches generated by ethylene alone mimic the commercial ethylene-propylene copolymers but possess tailored microstructures. Mechanistic insights by an in-depth DFT calculation on the selective branch pattern and distribution fully unravel the difference of nickel and palladium system. This work provides creative perspectives for chain walking polymerization and beyond, particularly shows how to generate, control, analyze, and predict branch in polymer synthesis. This work also for the first time demonstrates the precise synthesis of ethylene-propylene copolymer from ethylene alone at industrial temperatures.

## Methods

Methods and detailed experiments are provided in Supplementary Information.

## Data availability

The authors declare that all data supporting the findings of this study are available in the text or the Supplementary Materials. (1) Supplementary Information: 1) General synthesis and polymerization procedures, characterization and analysis data (NMR, DSC, XRD, and GPC) for ligand, nickel complex, and polymers. 2) DFT calculated details. Detailed mechanistic insights into nickel and palladium system. (2) Supplementary XYZ file for DFT calculation: The optimized Cartesian coordinates of all stationary points together with their single-point energy (a.u.) in solution. (3) Supplementary CIF file for ipty-Ni catalyst: The X-ray crystallographic data for ipty-Ni has been deposited at the Cambridge Crystallographic Data Center (CCDC) under the deposition number 1974098. These data can be obtained free of charge via www.ccdc.cam.ac.uk/data_request/cif. (4) Supplementary Excel file for statistic model: An Excel file including the statistic model for the calculation of branch degree, branch pattern, and branch distribution.

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

## Acknowledgements

We greatly appreciate Prof. Dr. Stefan Mecking from University of Konstanz for the discussion of branch formation and the statistical model of analyzing branch. We are also thankful for financial support from the National Natural Science Foundation of China (Nos. 22122110 and 21871250 for Z.J. and 22171038 for X.K.), the Jilin Provincial Science and Technology Department Program (No. 20200801009GH for Z.J.), and the Dalian Young Star of Science and Technology Project (No. 2019RQ88 for X.K.).

## Author contributions

Z.J. conceived the study and supervised this project. Y.Z. prepared all catalysts and performed polymerization experiments. X.K. performed DFT calculations and wrote the corresponding part in the manuscript. Z.J. wrote the manuscript. All authors have given approval to the final version of the manuscript.

## Competing interests

The authors declare no competing interests.
