## [Peer Review File · Nature Communications]

Selective branch formation in ethylene polymerization to access precise ethylene-propylene copolymersREVIEWER COMMENTS

Reviewer #1 (Remarks to the Author):

In this manuscript Jian et al. report on an insertion polymerization of ethylene to defined branched microstructures. Unlike known 'chain walking' ethylene polymerizations to branched polymers with random branch distributions, the selectivity for insertion into non-sterically encumbered 2-alkyls along with a high rate of chain walking results in defined branching patterns and very high degrees of branching. In other words, the extremely high chain walking rate is harnessed here by this unique selectivity to enable this defined structures. These microstructures are also appealing as they are comparable to commercial ethylene-propylene polymers, but can be generated from ethylene alone. These findings should clearly appeal to the readership of Nature Communications.

The manuscript requires major revisions to increase clarity, exemplary points to this end:

That the branch patterns follow a probability of $p(1-p)^n$ underlines and proves the validity of the mechanistic picture. However, to term the branch structure as 'programmable' is misleading. It's also rather an analysis of the branch statistics than a program.

Overall, a more stringent and somewhat condensed discussion of the branch patterns formed could improve readability.

Referring to an excel sheet in the last column of Table 3 appears unfinished.

In Figure 2, the chain growth steps from the 1,4 metal alkyl, 1,6 metal alkyl etc. that actually generate these branch motifs in the polymer should be indicated.

In this Figure and also the Figure in Table 2, indicating more prominently in the graphic that the 1,2 structure is not observed could also be helpful.

The palladium(II) catalyst precursor is shown in Scheme 2, but not mentioned anywhere in the text. Thus, when the branch patterns obtained with this catalyst are discussed, this comes a surprise ('These ultra-highly branched polyethylenes with exclusive methyl branches produced by the palladium precursor ipty-Pd were...').

The mechanistic discussion is rather lengthy, also some incongruencies appear like the conclusion that the reaction is kinetically controlled for the Ni catalyst but thermodynamically for the Pd catalyst. Both yield essentially the same branch pattern, so how should this be thermodynamically controlled? A focus on the origin of selectivity would be desirable. That the beta hydrogen elimination products/transition states could not be located appears awkward, as do the relative energies of some compounds (e.g. 8reins vs. 8reins-iso) and a critical consideration of the results appears advisable.

Reviewer #2 (Remarks to the Author):

Zhang, Y.; Kang, X.; and Jian, Z. have reported Ni and Pd (II) diimine catalysts capable of making branched polyethylene with near-exclusive methyl branching and high branch density. The resulting polymers contain high amounts of the 1,4-branching distribution. They have made a statistical model that is descriptive of the branching density achieved by these catalyst. The authors have also proposed a mechanism for branching supported by DFT calculations, suggesting two different methods of control kinetic (Nickel species) and thermodynamic (Palladium species).
Major Comments:

Overall, I think this work deserves to be published in Nature Communications, the results are new, the analysis is deep, and the concept is impactful. My biggest criticism with the manuscript is not related to science, but more to the way it is written. The DFT analysis takes up a lot of space within the document and while the DFT results are consistent with the experiment, I personally don't think it is necessary to provide the energy landscape for both nickel and palladium. I believe that detailing one and highlighting the differences would have been enough.

The other thing that I thought was missing is a mechanistic explanation of what the exclusive

formation of methyl branches meant. Here is what I wanted to see:

High degree of branching \diamond high frequency for b-H elimination

Exclusive formation of methyl branches \diamond 2,1 insertion of α -olefin predominant

High molecular weight formed \diamond intramolecular chain transfer is not occurring (α -olefin formed after b-H elimination is not replaced by ethylene)

High molecular weight formed \diamond 2,1 insertion is reversible but only to form terminal olefin.

I believe the DFT validates all these statement (and more) but I believe it is relevant to draw these conclusions from the polymerization data first.

I also did not like that the authors mention using a newly designed complex despite previously publishing the same ligand structure, and Pd catalyst (ref. 25)

Minor Comments:

Throughout the manuscript I made a few comments regarding the language use. This is minor but I am sharing it anyway.

"Over the past more than two decades, however, precise control on branches to produce well-defined polymer microstructures is a longstanding challenge, because of the uncontrolled chain walking process. A mixture of branch patterns including methyl branch and higher branches (C2, C3, C4, and C4+) predominantly occur (Scheme 1)."

While true, I think the work by Zhinbing Ye and Zhibing Guan should be cited here because they demonstrated that simple variation in ethylene pressure gives them control over the polymer's architecture.

"These events evidently inhibit the synthesis of ethylene-propylene elastomers from ethylene at industrial temperatures of 40 oC~70 oC."

I am surprised that industry would want polymerization temperatures as low as 40 oC. Maybe a citation to validate this claim would be beneficial

"To distinguish the branch pattern in these (ultra-)highly branched polyethylenes"

I find the use of the term "ultra-highly" to be excessive. I don't know what it means, is there a quantifiable difference between highly branched and ultra-highly branched?

The exclusive methyl branches result in a peculiarly clear and clean ¹³C NMR spectrum and thus allows for an in-depth analysis on branch distribution.

"Peculiarly clear and clean NMR" is not a good description, are the authors referring to a NMR with few peaks that were all sharp?

"(21.0%-20.8%)/21% = 1%), 6.1% of 1,8-Me₂ unit, 1.8% of 1,10-Me₂ unit, 0.5% of 1,12-Me₂"

I really like the statistical model but I find the calculation of the difference to be unnecessary. I don't need the authors to calculate the difference between 21.0% and 20.8%. The model is clearly good.

"programmable (herein we define 'programmable' as that ¹³C NMR data, branch number and distribution is predicted by a program simply, see Excel),"

I am not sold on the term programmable, because it is not like the authors have control over the branch number, they just have a model that allow them to determine the architecture of the polymer.

"It should be noted that such a high propylene content of 69.0 wt% (Table 2, entry 8) in the polymer produced by ethylene polymerization is prominent."

I did not understand this sentence.

Reviewer #3 (Remarks to the Author):

The paper reports on the synthesis of polyethylene with branching characteristics that are commonly achieved by a copolymerization of ethylene and propylene with solely methyl branches and a branch distribution of 1,4- Me₂ together with an ultra-high degree of branching. The authors provide a detailed study to show both the outcomes of ethylene polymerization with prepared ip₂-Ni and ip₂-Pd catalysts. The findings are excellent and demonstrate the unusual properties of the catalysts to give highly desired materials. The mechanistic investigations are detailed and explain the findings accurately.

However, throughout the text and title authors call their polymers 'elastomers' although a direct comparison of the prepared polymer and the commercially available one is never demonstrated. Either the authors show that indeed the similar branching characteristics lead in fact to the same elastic properties or if the authors do not want to demonstrate it, they should simply call these

polymers ethylene-propylene copolymers and leave the term 'elastomers' out as it is not proven and tested. Additionally, the term "programmable" is misleading and it is not clear what is 'programmable', as it gives the impression that the microstructure can be predetermined. Please use another term that makes clearer what is meant to be described.

Please correct the wording for some terms in the introduction such as:

"Selective branch formation thus is an extremely hard event", "largest scale synthetic polymers", "this is particularly more challenge".

The paper will be suitable for Nature Communication and should be accepted after minor corrections.

Reviewer: 1

Comments:

In this manuscript Jian et al. report on an insertion polymerization of ethylene to defined branched microstructures. Unlike known 'chain walking' ethylene polymerizations to branched polymers with random branch distributions, the selectivity for insertion into non-sterically encumbered 2-alkyls along with a high rate of chain walking results in defined branching patterns and very high degrees of branching. In other words, the extremely high chain walking rate is harnessed here by this unique selectivity to enable this defined structures. These microstructures are also appealing as they are comparable to commercial ethylene-propylene polymers, but can be generated from ethylene alone. **These findings should clearly appeal to the readership of Nature Communications.** The manuscript requires major revisions to increase clarity, exemplary points to this end:

1. That the branch patterns follow a probability of $p(1-p)^n$ underlines and proves the validity of the mechanistic picture. However, to term the branch structure as 'programmable' is misleading. It's also rather an analysis of the branch statistics than a program.

Answer: Thanks for this advice. Although we have defined the term of "programmable" in the text, we feel it is still misleading. Thus, in the Abstract and text, we have changed "thus far unsolved branch distribution is unprecedentedly programmable by using a simple program of $p(1-p)^n$ " to "thus far unsolved branch distribution is unprecedentedly predictable and computable by using a simple statistic of $p(1-p)^n$ ". Also, "programmable" is changed to "computable" when necessary.

2. Overall, a more stringent and somewhat condensed discussion of the branch patterns formed could improve readability.

Answer: We have revised the discussion of the branch patterns and have given a brief and condensed discussion. These following sentences have been deleted in the text and the detailed discussion has been moved into supplementary information:

"Also, methyl branches of 205/1000 C generated from ethylene polymerization signifies the formation of an ethylene-propylene copolymer with a greatly high content of propylene (61.5 wt%, Table 2, entry 7) using ethylene as the sole building block."

"the 1,8-Me₂ unit structure is formed by two growth steps and one branch formation [probability: $p(1-p)^2$], etc"

"Notably, this also suggests that branch formation is a well-behaved, single-site behavior from the viewpoint of the catalyst."

"(herein we define 'programmable' as that ¹³C NMR data, branch number and distribution is predicted by a program simply, see Excel)" "(the integration of all Me resonances at around 19.7 ppm is defined as 1.0)"

"(just play around the attached Excel)"

"It should be noted that such a high propylene content of 69.0 wt% (Table 2, entry 8) in the polymer produced by ethylene polymerization is prominent."

"To have a better understanding on above differences, the optimized geometrics for TS_{78ins}, TS_{78reins}, and

TS_{8BHE} were comparatively analyzed as shown in (see SI, Figure S2). In TS_{8BHE}, the bigger steric repulsion between MH and phenyl groups, as suggested by the bigger dihedral angles (126/129° vs. 122/121° in TS_{78ins} and 121/120° TS_{78reins}), can account for the less stability of TS_{8BHE} in comparison with TS_{78ins} and TS_{78reins}.”

*“And the relaxed scans (see SI, Figure S4) on C2–H1 distance and dihedral angle \angle H1-Ni-C2-C1 from TS_{78reins} to **6** direction indicate that neither a maximum nor a minimum point is found, thus further confirming our results.”*

“Further energy decomposition (See SI for computational details,^{50,51} and structural analyses (see SI, Figure S3) indicate that the less stability of TS_{910ins} can be ascribed to the bigger repulsion of poly(ethylene) chain/ethylene monomer and auxiliary ligand, as suggested by a bigger deformation energy ($\Delta E_{def} = \Delta E_{def(A)} + \Delta E_{def(B)} = 77.3$ vs. 55.5 kcal mol⁻¹ in TS_{910ins}), the bigger dihedral angles (\angle $\alpha_1\beta_1\gamma_1/\angle$ $\alpha_1\beta_2\gamma_2$: 124/123° in TS_{910ins} vs. 121/122° in TS_{910ins}) and a farther distance ($\gamma_1-\gamma_2 = 8.24$ Å in TS_{910ins} vs. 7.96 Å in TS_{910ins}) between two phenyl centers of ligand.”

3. Referring to an excel sheet in the last column of Table 3 appears unfinished.

Answer: Thanks for this advice. We feel that the use of “1,(2n+4)-...” in the last column of Table 3 may be misleading. Now, “1,(2n+4)-...” is changed to “1,(2n+4)- [n ≥ 5]”.

4. In Figure 2, the chain growth steps from the 1,4 metal alkyl, 1,6 metal alkyl etc. that actually generate these branch motifs in the polymer should be indicated.

Answer: Thanks for this advice. We fully agree that the chain growth steps from the 1,4-metal alkyl actually generate these branch motifs, which are now indicated in Figure 4 (the previous Figure 2). Meanwhile, we also re-draw this figure.

5. In this Figure and also the Figure in Table 2, indicating more prominently in the graphic that the 1,2 structure is not observed could also be helpful.

Answer: It is a good advice. We have highlighted the ‘not observed 1,2- unit’ in Figures 3 and 4 and Table 2.

6. The palladium(II) catalyst precursor is shown in Scheme 2, but not mentioned anywhere in the text. Thus, when the branch patterns obtained with this catalyst are discussed, this comes a surprise (,These ultra-highly branched polyethylenes with exclusive methyl branches produced by the palladium precursor ip₂ty-Pd were...’).

*Answer: In the section of “Catalyst Design and Ethylene Polymerization.”, we have added “As a comparison, **ip₂ty-Pd** reported by us was also prepared (Figure 2),²⁸ which produced highly branched polyethylenes with unsolved microstructures at low temperatures of < 30 °C as well.”.*

7. The mechanistic discussion is rather lengthy, also some incongruencies appear like the conclusion that the reaction is kinetically controlled for the Ni catalyst but thermodynamically for the Pd catalyst. Both

yield essentially the same branch pattern, so how should this be thermodynamically controlled? A focus on the origin of selectivity would be desirable. That the beta hydrogen elimination products/transition states could not be located appears awkward, as do the relative energies of some compounds (e.g. δ_{reins} vs. $\delta_{\text{reins-iso}}$) and a critical consideration of the results appears advisable.

Answer:

(1) We also feel the mechanistic discussion is rather lengthy and substantially revise the section. We focus on selective branch formation of the nickel system, move the palladium system to SI and only give a comparison on difference. Based on the difference, the description in Abstract and Conclusion has been revised. We also delete the simple description of “the reaction is kinetically controlled for the Ni catalyst but thermodynamically for the Pd catalyst”.

(2) The origin of selectivity has been elucidated by the comparison on difference of nickel and palladium species. We have added one new paragraph and also shown this difference on the TOC:

Mechanistic Insight into Branch Formation (Difference of Nickel and Palladium System). The entire discussion and the related supplementary Notes and Figs. 3 and 10 can be found in Supplementary Information. Herein, we only focus on the difference of the origin of selectivity based on the same unique ligand. Inaccessible 1,2-Me₂ unit (distribution): (**Ni**) the unfavorable kinetics (+3.9 kcal mol⁻¹) and thermodynamics (+5.0 kcal mol⁻¹) and high barrier (21.9 kcal mol⁻¹) of subsequent ethylene insertion; (**Pd**) the favorable kinetics (-5.8 kcal mol⁻¹) but both unfavorable thermodynamics (+2.3 kcal mol⁻¹) and high barrier (17.8 kcal mol⁻¹) of subsequent ethylene insertion. Inaccessible 1,3-Me/Et unit (longer branches): (**Ni**) the extremely unfavorable kinetics (+14.0 kcal mol⁻¹); (**Pd**) the slightly unfavorable kinetics (+2.2 kcal mol⁻¹), indicating an impossibility at low temperature but a possibility at higher temperature. Available 1,4-Me₂ unit: (**Ni**) comparable barrier of isomerization (chain walking) with the precursor of 1,6-Me₂ unit (13.9 vs 13.1 kcal mol⁻¹) but lower barrier (10.3 kcal mol⁻¹) of subsequent ethylene insertion to form 1,4-Me₂ unit (**10**); (**Pd**) easy isomerization (chain walking) (7.3 kcal mol⁻¹) and comparable barrier (15.0 vs 15.5 kcal mol⁻¹) of subsequent ethylene insertion to form 1,4-Me₂ unit (**10**).

(3) The compounds δ_{reins} and $\delta_{\text{reins-iso}}$ are the isomer with different β -agostic-H types, which actually give the same 1,4-Me₂ unit. We have checked again for the relative energies and revised the description “ δ_{reins} (-29.7 kcal mol⁻¹, the β -agostic H from CH₃ of methyl branches) isomerizes to a more stable $\delta_{\text{reins-iso}}$ (-33.3 kcal mol⁻¹, the β -agostic H from CH₂ of backbone, also 1,4-Me₂ unit).”.

(4) We have re-calculated these beta-H elimination products $7_{\text{BHE-C}}$ and $8_{\text{BHE-C}}$ and re-checked the entire calculation. We have added detailed evidences and discussions for the not located intermediates to Supplementary Information (Supplementary Notes, Figs. 5-7). Taking $7_{\text{BHE-C}}$ as an example:

Supplementary Figure 5. Relaxed scan of potential energy surface (PES) for the distance (angstrom) of C2 and H1 and dihedral angle (degree) \angle H1-Ni-C2-C1 in **TS_{78reins}** at the DFT/BSI level.

Supplementary Figure 6. IRC analysis for **TS_{78reins}** and optimized processes of the **6** and **8_{reins}**.

As shown in supplementary Fig. 5, we can observe that the dihedral angle (degree) $\angle H1-Ni-C2-C1$ in $TS_{78reins}$ is about 60° and the $TS_{78reins}$ from its' geometry is very near to δ_{reins} . So we just scan the distance of C2 and H1 and dihedral angle $\angle H1-Ni-C2-C1$ in $TS_{78reins}$ to the direction of **6**, and no other transition state and intermediate are found (neither a maximum nor a minimum point is found). In addition, we also confirm that all transition state structures are shown to connect the reactant and product on either side via intrinsic reaction coordinate (IRC) following. For example, the structure $TS_{78reins}$ definitely connecting the reactant **6** and product δ_{reins} is verified (supplementary Fig. 6).

Reviewer: 2

Comments:

Zhang, Y.; Kang, X.; and Jian, Z. have reported Ni and Pd (II) diimine catalysts capable of making branched polyethylene with near-exclusive methyl branching and high branch density. The resulting polymers contain high amounts of the 1,4-branching distribution. They have made a statistical model that is descriptive of the branching density achieved by these catalyst. The authors have also proposed a mechanism for branching supported by DFT calculations, suggesting two different methods of control kinetic (Nickel species) and thermodynamic (Palladium species).

Major Comments:

1. Overall, I think this work deserves to be published in Nature Communications, the results are new, the analysis is deep, and the concept is impactful. My biggest criticism with the manuscript is not related to science, but more to the way it is written. The DFT analysis takes up a lot of space within the document and while the DFT results are consistent with the experiment, I personally don't think it is necessary to provide the energy landscape for both nickel and palladium. I believe that detailing one and highlighting the differences would have been enough.

Answer: We agree it. We also feel the mechanistic discussion is rather lengthy and majorly revise the section. We focus on selective branch formation of the nickel system and also shorten this section, move the palladium system to SI and only give a comparison on difference.

2. The other thing that I thought was missing is a mechanistic explanation of what the exclusive formation of methyl branches meant. Here is what I wanted to see:

High degree of branching \diamond high frequency for b-H elimination

Exclusive formation of methyl branches \diamond 2,1 insertion of α -olefin predominant

High molecular weight formed \diamond intramolecular chain transfer is not occurring (α -olefin formed after b-H elimination is not replaced by ethylene)

High molecular weight formed \diamond 2,1 insertion is reversible but only to form terminal olefin.

I believe the DFT validates all these statement (and more) but I believe it is relevant to draw these conclusions from the polymerization data first.

Answer: Thanks for this advice. In the section of Exclusive Branch Pattern and Selective Branch Distribution We have added the discussion "Based on pioneering insights,^{4,12-14} high molecular weight

of the obtained polyethylene implies the suppression of chain transfer that involves the replacement of the formed α -olefin by incoming ethylene. High degree of branching is attributed to high frequency of β -H elimination and subsequent re-insertion. Notably, the key exclusive formation of methyl branches means that 2,1-insertion of α -olefins predominates and no event occurs from 2,1-insertion of internal olefin that generates longer branches.”

3. I also did not like that the authors mention using a newly designed complex despite previously publishing the same ligand structure, and Pd catalyst (ref. 25)

Answer: We have changed “the newly-designed low-cost nickel(II) catalyst” to “the low-cost nickel(II) catalyst” in the last paragraph of Introduction.

Minor Comments:

Throughout the manuscript I made a few comments regarding the language use. This is minor but I am sharing it anyway.

4. “Over the past more than two decades, however, precise control on branches to produce well-defined polymer microstructures is a longstanding challenge, because of the uncontrolled chain walking process. A mixture of branch patterns including methyl branch and higher branches (C2, C3, C4, and C4+) predominantly occur (Scheme 1).” While true, I think the work by Zhibing Ye and Zhibing Guan should be cited here because they demonstrated that simple variation in ethylene pressure gives them control over the polymer’s architecture.

Answer: Thanks for this advice. Ref. 4 (Science 283, 2059-2062, 1999) from Prof. Zhibin Guan is related to this topic. In the Introduction, we have added “For instance, the architecture of polyethylene could readily be adjusted by a simple variation of ethylene pressure.^{4,24-26}” and also cited these three Refs. 24-26 from Prof. Zhibin Guan and Zhibin Ye (Polym. Chem. 2012, 3, 286; Chem. Commun. 2013, 49, 6235; Chem. Asian J. 2010, 5, 1058).

5. “These events evidently inhibit the synthesis of ethylene-propylene elastomers from ethylene at industrial temperatures of 40 oC~70 oC.” I am surprised that industry would want polymerization temperatures as low as 40 oC. Maybe a citation to validate this claim would be beneficial

Answer: The reaction temperature of 40 °C~70 °C was claimed in an international patent (Applicant: FASTECH S.R.L. Italy) that involves the preparation of EP copolymers. We have added it as Ref. 27 (WO2014/202715 A1).

6. “To distinguish the branch pattern in these (ultra-)highly branched polyethylenes” I find the use of the term “ultra-highly” to be excessive. I don’t know what it means, is there a quantifiable difference between highly branched and ultra-highly branched?

Answer: Thanks for this advice. Generally, PE with branching numbers of 75~120/1000C can be called as ‘highly’ branched PE (Eur. Polym. J. 2021, 142, 110100). In our case, the highest branching number reaches 230/1000C, almost two times higher. To highlight it, we use the term of “ultra-highly” branched

PE, which is similar to UHMWPE. However, we agree that the use of “ultra-highly” is slightly excessive, thus we have changed “ultra-highly” to “highly” in the section of Results and only retained it in the key sections of Abstract and Conclusion to highlight the difference.

7. The exclusive methyl branches result in a peculiarly clear and clean ^{13}C NMR spectrum and thus allows for an in-depth analysis on branch distribution. “Peculiarly clear and clean NMR” is not a good description, are the authors referring to a NMR with few peaks that were all sharp?

Answer: We have changed this sentence to “The exclusive methyl branches further allow for an in-depth analysis on branch distribution via ^{13}C NMR spectra.³⁶⁻³⁹”.

8. “(21.0%-20.8%)/21% = 1%), 6.1% of 1,8-Me₂ unit, 1.8% of 1,10-Me₂ unit, 0.5% of 1,12-Me₂” I really like the statistical model but I find the calculation of the difference to be unnecessary. I don’t need the authors to calculate the difference between 21.0% and 20.8%. The model is clearly good.

Answer: Thanks for this advice. We have changed “(21.0% from NMR data; thus only a deviation of 1%, calculated by (21.0%-20.8%)/21% = 1%)” to “(21.0% from NMR data)”.

9. “programmable (herein we define ‘programmable’ as that ^{13}C NMR data, branch number and distribution is predicted by a program simply, see Excel),” I am not sold on the term programmable, because it is not like the authors have control over the branch number, they just have a model that allow them to determine the architecture of the polymer.

Answer: Thanks for this advice. Although we have defined the term of “programmable” in the text, we feel it is still misleading. Thus, “programmable” is changed to “computable” when necessary.

10. “It should be noted that such a high propylene content of 69.0 wt% (Table 2, entry 8) in the polymer produced by ethylene polymerization is prominent.” I did not understand this sentence.

Answer: We have deleted this sentence to avoid a misunderstanding.

Reviewer: 3

Comments:

The paper reports on the synthesis of polyethylene with branching characteristics that are commonly achieved by a copolymerization of ethylene and propylene with solely methyl branches and a branch distribution of 1,4- Me₂ together with an ultra-high degree of branching. The authors provide a detailed study to show both the outcomes of ethylene polymerization with prepared ip₂ty-Ni and ip₂ty-Pd catalysts. The findings are excellent and demonstrate the unusual properties of the catalysts to give highly desired materials. The mechanistic investigations are detailed and explain the findings accurately. **The paper will be suitable for Nature Communication and should be accepted after minor corrections.**

1. However, throughout the text and title authors call their polymers ‘elastomers’ although a direct comparison of the prepared polymer and the commercially available one is never demonstrated. Either

the authors show that indeed the similar branching characteristics lead in fact to the same elastic properties or if the authors do not want to demonstrate it, they should simply call these polymers ethylene-propylene copolymers and leave the term ‘elastomers’ out as it is not proven and tested.

Answer: We fully agree it. We have changed “elastomers” to “copolymers”.

2. Additionally, the term “programmable” is misleading and it is not clear what is ‘programmable’, as it gives the impression that the microstructure can be predetermined. Please use another term that makes clearer what is meant to be described.

Answer: Thanks for this advice. Although we have defined the term of “programmable” in the text, we feel it is still misleading. Thus, “programmable” is changed to “computable” when necessary.

3. Please correct the wording for some terms in the introduction such as: “Selective branch formation thus is an extremely hard event”, “largest scale synthetic polymers”, “this is particularly more challenge”.

Answer: The terms have been corrected. “Selective branch formation thus is an extremely hard event” changes to “Therefore, selective branch formation is difficult”; “largest scale synthetic polymers” changes to “the most important polymer by scale”; “this is particularly more challenge” to “this is particularly challenging”.

In addition, throughout the text we have also revised some other misleading descriptions with highlights.

REVIEWERS' COMMENTS

Reviewer #1 (Remarks to the Author):

The authors have addressed all major points satisfactorily.
I would suggest to revise the colloquial expression '(also play around the other p value in supplementary Excel).'

Reviewer #2 (Remarks to the Author):

The authors addressed of my comments and i am now satisfied with the manuscript.

Reviewer: 1

Comments:

The authors have addressed all major points satisfactorily.

I would suggest to revise the colloquial expression '(also play around the other p value in supplementary Excel).'

Answer: Thanks. 'play around' has been changed to 'input'.

Reviewer: 2

Comments:

The authors addressed of my comments and I am now satisfied with the manuscript.

Answer: Thanks.